# Description of a Cohort with a New Truncating *MYBPC3* Variant for Hypertrophic Cardiomyopathy in Northern Spain

**DOI:** 10.3390/genes14040840

**Published:** 2023-03-30

**Authors:** Natalia Fernández Suárez, María Teresa Viadero Ubierna, Jesús Garde Basas, María Esther Onecha de la Fuente, María Teresa Amigo Lanza, Gonzalo Martin Gorria, Adrián Rivas Pérez, Luis Ruiz Guerrero, Domingo González-Lamuño

**Affiliations:** 1Pediatric Cardiology Division, Pediatric Department, “Marqués de Valdecilla” University Hospital, 39008 Santander, Spain; natalia.fernandezs@scsalud.es (N.F.S.); mteresa.viadero@scsalud.es (M.T.V.U.); jesus.garde@scsalud.es (J.G.B.); 2Cardiovascular Genetic Disease Division, Molecular Genetic Department, “Marqués de Valdecilla” University Hospital, 39008 Santander, Spain; 3Pediatric Laboratory, Department of Medical and Surgical Sciences, Faculty of Medicine, University of Cantabria, 39011 Santander, Spain; 4Familial Heart Disease Division, Cardioloy Department, “Marqués de Valdecilla” University Hospital, 39008 Santander, Spain; gonzalo.martin@scsalud.es (G.M.G.); cardiofamiliar.humv@scsalud.es (L.R.G.); 5Pediatric Nephrology, Metabolism and Genetic Disease Division, Pediatric Department, “Marqués de Valdecilla” University Hospital, 39008 Santander, Spain; 6Research Institute Valdecilla (IDIVAL), 39011 Santander, Spain

**Keywords:** hypertrophic cardiomyopathy, *MYBPC3*, genotype-phenotype, sarcomeric gene variant

## Abstract

Background: The pathogenicity of the different genetic variants causing hypertrophic cardiomyopathy (HCM) and the genotype/phenotype correlations are difficult to assess in clinical practice, as most mutations are unique or identified in non-informative families. Pathogenic variants in the sarcomeric gene *MYBPC3* inherited with an autosomal dominant pattern, whereas incomplete and age-dependent penetrance are the most common causes of HCM. Methods: We describe the clinical characteristics of a new truncating *MYBPC3* variant, p.Val931Glyfs*120, in 75 subjects from 18 different families from northern Spain with the p.Val931Glyfs*120 variant. Results: Our cohort allows us to estimate the penetrance and prognosis of this variant. The penetrance of the disease increases with age, whereas 50% of males in our sample developed HCM by the age of 36 years old, and 50% of women developed the disease by the time they reached 48 years of age (*p* = 0.104). Men have more documented arrhythmias with potential risk of sudden death (*p* = 0.018), requiring implantation of cardioverter defibrillators (*p* = 0.024). Semi-professional/competitive sport among males is related to earlier onset of HCM (*p* = 0.004). Conclusions: The p.Val931Glyfs*120 truncating variant in *MYBPC3* is associated with a moderate phenotype of HCM, with a high penetrance, onset in middle age, and a worse outcome in males due to higher risk of sudden death due to arrhythmias.

## 1. Introduction

Hypertrophic cardiomyopathy (HCM) is a disease prevalent in our setting (1/500 individuals) that is genetic based (generally with autosomal dominant inheritance) and inherited through families [1,2]. There are more than 1000 known variants in the genes that code for cardiac sarcomeric proteins, which alter the functionality of the cardiac sarcomere, generating an increase in muscle mass and myofibrillar disarray [3]. This leads to an obstruction in the outflow tract of the left ventricle, coronary ischemia due to narrowing of the small blood vessels, and severe arrhythmias [4], and is the main cause of sudden death of cardiac origin in children and young people. The existing treatments (β-blockers, myomectomy, ICD, transplant, etc.) are palliative [5].

Of all known variants, 42% of them affect the *MYH7* gene (myosin heavy chain), 48% affect the *MYBPC3* gene (myosin-binding protein C), and the remaining 10% affect other sarcomeric proteins [3,6]. In 2018, the Heart Failure Society of America Practice Guideline recommended offering a genetic study to all patients with a clinical diagnosis of HCM, particularly if there was a family history of the disease [7]. Genotype-phenotype association studies have shown a degree of variable penetrance (50–70%) for many variants, while the influence of other endogenous or environmental factors on the age of onset and severity of the disease has not been well established [8]. Furthermore, most pathogenic variants are limited to a small number of families or even specific individuals [9]. Therefore, the study of recently identified pathogenic variants with a founder effect that affect homogenous populations represents a great opportunity for the description of clinical phenotypes [10].

For the *MYBPC3* gene, more than 150 variants associated with HCM have been described [11]; 70% of them are truncating variants [6]. Several studies have attempted to define the phenotype associated with the variant in question [12], and although the first papers initially pointed towards a later onset, less hypertrophy, and a better outcome [13,14], more recent studies indicate that they are no different from patients with variants in thick and thin filament genes [12,15,16]. The importance of establishing genotype-phenotype correlations lies in the fact that this would enable improved risk stratification and changes in the management of carriers [17].

The main objective of this study is to describe the clinical phenotype, penetrance, and prognosis of a new *MYBPC3* variant, p.Val931Glyfs*120, present in several families from northern Spain (Cantabria), in order to adapt the monitoring and advice given to these patients. To date, no descriptions relating to the clinical impact of this mutation have been reported.

## 2. Materials and Methods

The study included 108 individuals from 18 unrelated Cantabrian families with HCM and carriers of the p.Val931Glyfs*120 *MYBPC3* gene variant (NM_000256.3:c.2791_2792insG).

The recruited participants were being monitored by the Pediatric Cardiology and Family Heart Disease Units of the “Marqués de Valdecilla” University Hospital.

All 18 index cases (13 males, 5 females) were recruited from the Cardiology Units of the “Marqués de Valdecilla” University Hospital, where 13 patients (72.2%; 9 males, 4 females) attended for cardiac symptoms (dyspnea, chest pain, syncope and/or palpitations); one male (5.5%) was referred after a cardiorespiratory arrest (CRA) and 4 patients (22.2%; 3 males, 1 female) were diagnosed from an ECG suspicious for HCM performed in a routine health check-up. From these cases, a pedigree was designed for each case and a genetic study was offered to 90 first- to third-degree relatives [7]. The DNA was extracted using standard peripheral blood sample protocols. Probands were studied by Next generation sequencing (NGS) using an 18/118 gene panel at the Health in Code^®^ laboratory in A Coruña, and family members were studied by Sanger sequencing of the variants detected in the proband patient (Appendix A). They were all duly informed. The adults signed an informed consent form, whereas the minors gave their consent verbally, with their parents/guardians signing the written consent document. The study has the approval of the Cantabria Clinical Research Ethics Committee (CEIC-Idival, Code 2018.286; 15 February 2019).

All the affected subjects and the carrying relatives are currently being monitored by the Cardiology Units of our hospital. A medical history was produced for all the carriers, and they were all given a physical examination, ECG, and echocardiogram, according to the latest HCM guidelines published by the European Society of Cardiology [18]. According to the monitoring protocols and the outcome of each patient, magnetic resonance imaging (MRI) studies were carried out.

The diagnosis of HCM based on the aforementioned Guides [18] included the following criteria: maximum thickness of the left ventricle wall ≥15 mm in adult index patients, ≥13 mm in adult relatives, or >2 SD in children in the absence of any other disease that could cause left ventricular hypertrophy. The patients underwent risk stratification and were managed in accordance with the latest recommendations for this disease [19].

For the statistical analysis, a database was created (Excel 2000, Microsoft) and the statistical program IBM SPSS for PC (version 28.0.1) was used. Parametric tests were performed (Student’s *t*-test and ANOVA) for a comparison of independent sample means where the distribution permitted (Levene’s test for homogeneity of variance). Non-parametric tests were used (Mann–Whitney U-test and Kruskal–Wallis test), where the distribution did not meet the above-mentioned criterion. Independence between the two qualitative variables was tested using the chi-square test. The log-rank test was used to compare the disease-free survival between the sexes. Statistically significant differences were considered where *p* < 0.05.

## 3. Results

The study included 108 subjects: relatives of 18 index cases from 18 non-related first grade families, carriers of the p.Val931Glyfs*120 *MYBPC3* gene variant, and classified as pathogenic following ACMG criteria [20]. We identified 75 subjects carrying the variant and 33 non-carrier relatives (Figure 1).

In 67 cases (89.3%), the p.Val931Glyfs*120 variant in *MYBPC3* was identified, whereas in 8 cases (10.6%), the genetic analyses revealed an association with a second variant related to the development of HCM, in the sarcomeric genes *MYL2*, *MYH6*, or *TNNT2* or in the protein-coding gene *RBM20* [21,22]. The variants in the *MYL2*, *MYH6*, and *RBM20* (Appendix C) genes present in 6 subjects were classified as of uncertain/unknown significance, whereas the p.Arg286His variant in the *TNNT2* (Appendix C) gene present in two individuals from the same family was classified as pathogenic [23]. No significant differences were found in the incidence of HCM among carriers of one or two variants (*p* = 0.853). Moreover, three relatives (3.3%) inherited only the variants of uncertain significance in *RBM20* and *MYL2*, and their cardiac phenotypes were normal.

The clinical assessment confirmed the presence of HCM in 44 of the 75 subjects carrying the p.Val931Glyfs*120 variant (58.7%), while one patient (2.2%) developed dilated cardiomyopathy (DCM) and ischemic cardiomyopathy, and was therefore excluded from the statistical analysis. Five of the patients under study died at a relatively early age (60, 62, 64, 70, and 77 years old), three of them due to heart failure as an advanced stage of HCM (Figure 2).

In the group of non-carrier relatives, an ECG was performed on all of them and an echocardiogram on 25 of the 33 (75.7%); both studies were normal in all cases.

An ECG and echocardiogram were performed on all the carriers. In this group, in 44 patients (16 women and 28 men), the echocardiograms were consistent with HCM according to the latest Guidelines [18]. The average age of the carriers with HCM was 55 ± 17 years, whereas in the carriers with a negative phenotype, that is to say clinically asymptomatic and with a normal echocardiography, the average age was 26 ± 15 years (*p* < 0.001). Additionally, 45.7% of female carriers had developed the disease at the time of the study (5 years) compared with 71.8% of the male carriers (*p* = 0.009). No significant differences were found in the current average age of the men and women with a positive phenotype (Table 1, Appendix B). However, there is an almost significant difference (*p* = 0.063) in the current average age of males and females with a negative phenotype (20 ± 14 years and 30 ± 14 years, respectively), being higher in the women who have not yet developed the disease.

The penetrance of the disease increases with age, meaning that 50% of the men in our sample developed HCM at the approximate age of 36 years old, while 50% of the women developed the disease at the age of 48 years old. These differences are not statistically significant (*p* = 0.104) (Figure 3). The average age at diagnosis was 40 ± 16 years old, with the youngest patient being diagnosed at 3 years of age and the oldest at 69 years of age. We did not find any significant differences between the sexes in this regard (Table 1, Appendix B).

Regarding the reason for the cardiological examination of our patients, 45.5% was due to the presence of suspected symptoms of heart disease (breathlessness, chest pain, palpitations), followed by a family history of HCM (43.2%) and, finally, asymptomatic patients, where possible signs of HCM were revealed by an ECG (11.4%). Among the men, the predominant reason was an examination due to symptoms (50%), whereas the women were generally referred to the cardiologist because they were relatives of patients with HCM (56.3%), although the difference is not significant (*p* = 0.382) (Table 1; Appendix B). In this regard, it should be noted that in 50% of the families, there were one or more cases of SCD in persons not under study (Figure 1).

In our sample of patients with HCM, we found no significant differences between sexes respecting the average thickness of the interventricular septum (IVS) (*p* = 0.208), nor the average thickness of the left ventricle posterior wall (LVPW) (*p* = 0.759), although in both cases, women had slightly lower thicknesses than males. Regarding the type of hypertrophy, asymmetric forms with septum impairment were predominant in both sexes. We did not observe any significant differences in the average thickness of the IVS in carriers with a negative phenotype or in the average thickness of the LVPW (*p* = 0.802 and *p* = 0.632, respectively). However, as was to be expected, there was a clear difference in IVS thickness between carriers with a positive and negative phenotype (*p* < 0.001), and the same is also true of the LVPW values (*p* = 0.001).

Regarding the electrocardiogram findings, three carriers with a negative phenotype (10.0%) had irregularities in their ECG that were consistent with HCM (high voltages in left precordial leads, left axis, and negative T wave in III, and/or negative T waves in left precordial leads). These irregularities were evident in 37 of the subjects with HCM (84.1%), and only three relatives with an echocardiographic diagnosis of HCM had a normal ECG (6.8%). Therefore, there are significant differences between carriers with a developed phenotype and those with a negative phenotype in the ECG (*p* < 0.001).

A total of 54.5% of the patients with a positive phenotype had symptoms, the most common of which was breathlessness. No significant differences were recorded between the sexes in this regard (*p* = 0.647). Most of the patients were in the NYHA class I-II (75%), without any significant differences between the sexes (*p* = 0.897).

A total of 25% of patients of both sexes developed hypertrophic obstructive cardiomyopathy (HOCM); 15.9% showed signs of systolic dysfunction, and 75% diastolic dysfunction in the echocardiogram. Both are more frequent in males, but the differences are not significant (Table 1, Appendix B).

If the left atrium is dilated when the diameter is ≥45 mm [24], we observed that 50% of patients had dilatation of the left atrium. Consequently, 28.6% of the sick males and 43.8% of the sick females (*p* = 0.307) had atrial fibrillation (AF).

Moreover, although more than 60% of patients of both sexes had arrhythmias, the episodes of non-sustained ventricular tachycardia (NSVT) detected in Holter monitoring were significantly more common in men (*p* = 0.018).

A cardiac MRI was performed on 63.6% of the patients diagnosed with HCM, all being consistent with the disease; 70.6% of the sick men showed signs of fibrosis, compared with 36.4% of the sick women (*p* = 0.074).

On the other hand, 53.6% of the sick men had an ICD compared with 18.8% of the sick women (*p* = 0.024). In most cases, an ICD was recommended for primary prevention and only two patients (11.1%) received appropriate shocks.

A total of 61.4% of patients were receiving drug treatment. Two patients required a heart transplant, three a Morrow myomectomy, and one an alcohol septal ablation.

Nine patients suffered events, with no significant differences detected between the sexes (*p* = 0.394). Three patients died from heart failure and another two from causes unrelated to this disease.

A total of 25% of patients played sport at semi-professional/competitive levels prior to diagnosis, compared with 10.5% of the carriers who still have a negative phenotype (*p* = 0.870). Regarding the men, 39.3% of the patients played sport at this level, compared with 45.5% of the carriers with a negative phenotype (*p* = 0.165). Moreover, the average onset age of the sick men who played high-level sport was 29 ± 8 years, compared with those who did not play sport, with an average onset age of 45 ± 11 years, resulting in a statistically significant difference (*p* < 0.001). None of the sick women previously played sport at this level, and only two who still have a negative phenotype play sport.

## 4. Discussion

As far as we are aware, this is the first study to describe the clinical phenotype of the p.Val931Glyfs*120 *MYBPC3* gene variant, demonstrating co-segregation in 18 not-related families affected by HCM.

This variant was first referred to in Gómez et al. (2014) as the cause of hypertrophic cardiomyopathy [25]. However, this is the first study to describe the genotype-phenotype of a cohort for this variant.

The pathogenic mechanism of the p.Val931Glyfs* variant would be by truncation, given that the insertion of a guanine would produce a change in the translation of a valine by a glycine at position 931, with a change in the reading frame that would generate a premature stop codon 120 amino acids downstream. Aberrant translations could be degraded by the cellular machinery without being translated, or could do so, giving rise to non-functional truncated peptides [26]. The purpose of the MYBPC3 protein is to anchor the myosin to the sarcomere A-band, providing stability during muscle contraction; the disease would therefore occur through haploinsufficiency [27] (Figure 4).

As described in most series with truncating *MYBPC3* variants, the penetrance for this variant increases with age (*p* < 0.001) [26,28]. In our cohort, there are no significant differences between the sexes in terms of the disease onset age or the penetrance (Figure 3). This is in line with the results of other Spanish and international studies carried out in large populations of patients with HCM [29,30,31]. However, other published cohorts have found significant differences in the onset age, with men being younger at diagnosis [10,31,32]. We postulate a possible protective role of endocrine factors in women [10] or a lack of attention to early signs of disease in women and the absence of any special recommendation for screening programs [32]. In fact, we found an almost significant difference (*p* = 0.063) in the current average age of males and women with a negative phenotype, with women who are yet to develop the disease being older. In our case, the lack of a bigger difference between the sexes could be since we carried out an active search for relatives carrying the disease, both with symptoms and asymptomatic, irrespective of their sex.

The predominant clinical presentation in our population is moderate HCM with septal predominance, limited obstruction, but a high percentage of arrhythmias. These characteristics are like those described for other variants in the *MYBPC3* gene that lead to truncation of the protein [12,17,28,33].

One of the carrying relatives developed a phenotype consistent with dilated cardiomyopathy. It is possible that they may be in an advanced phase of hypertrophic cardiomyopathy, where dilatation and systolic dysfunction are predominant, or that this is the initial manifestation. These findings had already been previously described in other articles as the final evolving forms of HCM [12,34].

Mention should be made of the fact that in the cardiac MRI of the men with HCM, more fibrosis was observed than in women. This would explain the higher number of episodes of NSVT in the Holter monitoring of the men. We consider this one of the main prognostic differences between the sexes. In fact, these findings would be in line with the fact that the percentage of male patients who have an ICD in our sample is much higher than that of women.

In our cohort, there are no significant differences regarding symptoms by sex. However, in the study by Sabater et al. [28], female patients had more breathlessness and chest pain. This could be because in our study, younger people have been diagnosed in earlier phases of the disease.

As regards the advice given to young male carriers on lifestyle and sport, we find it interesting to note that there is a significant difference between the age of onset of patients who practice any sport at a semi-professional/competitive level and those who do not, being the lowest age in the first group. The practice of intense exercise on a regular basis generates certain physiological changes in the heart in “healthy” subjects, including an increase in the mass and volume of the left ventricle [35]. Pathological studies of hearts with HCM show increased left ventricular muscle mass at the expense of hypertrophy of cardiac myocytes, with decreased ventricular volume, as well as myofibrillar disorganization and coronary small vessel disease [18]. It has been shown that intense exercise increases the gradient through the left ventricular outflow tract (the obstruction increases), and shortens diastolic filling time; it can cause cardiac ischemia due to a mismatch between demand and supply and small vessel disease, and increases the risk of atrial and ventricular arrhythmias due to sympathetic activation [36]. For this reason, it has traditionally been considered that competitive sport was a factor which could accelerate the advance of HCM in genetically predisposed subjects and even contribute to the development of malignant arrhythmias. The Guidelines, in the past, were very restrictive in respect of the practice of competitive sport [37,38].

More recent studies, in both transgenic mice and in humans, have not only questioned this but also observed a better outcome in patients who play sport, with less cardiac wall hypertrophy and less systolic dysfunction in the outcome [39,40,41].

Taking a detailed three-generation family history, we were able to estimate penetrance for this familial condition and estimate the impact of some aspects as physical activity. Although we did not observe a statistically significant difference, according to the long-rank graph, it is remarkable that at the age of 50 years, 25% of men remain with a negative phenotype, while at the same age, 46% of women remain with a normal echocardiography. The estimate of penetrance has some limitations. The data were generated assuming a simple copula model, and actual familial disease data may exhibit more complex patterns of residual disease correlation so they can be extrapolated to the general population.

Our cohort shared both variants and environment, thus some interactions between them can be assessed. According to our study, the p.Val931Glyfs*120 *MYBPC3* variant is clearly pathogenic, with a strong founder effect in the northern Spanish region of Cantabria, where all the families are from. In addition, the surnames of several affected families are of very specific origin (Pasiego origin). The Pasiegos Valleys have traditionally been considered a “closed population with strong endogamy” [42]. This possible endogamy has been proven in numerous studies analyzing the polymorphic genetic markers of both the HLA system and the mitochondrial DNA or the Y chromosome. Moreover, Cardoso et al. [43] observed a low diversity of female lineages through an analysis of the mitochondrial DNA of a group of individuals of Pasiego origin with no maternal lineage. The founder effect would reduce the amount of existing genetic variability within our population, and as a result of this, certain phenotypes or particular genes might be highlighted in this population, such as the p.Val931Glyfs*120 *MYBPC3* variant.

## 5. Limitations

As is the case in most studies of genotype-phenotype correlation in HCM, our study is also limited by the small size of the families and the scarcity of families with identical variants [15,44]. In addition, in our cohort, some families present some other mutation in sarcomeric genes or genes related to cardiomyopathy, which could produce a synergy effect with the variant under study. Furthermore, the penetrance data is only applicable to our sample, since the variant frequency has not been tested in a local population. On the other hand, our study is both retrospective and prospective, with a follow-up time from its beginning of 5 years. Future studies with larger cohorts will therefore be necessary to help to clarify the genotype-phenotype relations for this pathogenic variant.

## 6. Conclusions

According to the results of our cohort, the pathogenic variant Val931Glyfs*120 has a phenotype pattern like that described for other *MYBPC3* gene variants with a more advanced onset age, a more benign clinical course, and a lower incidence of SCD in comparison with variants affecting the *MYH7* gene. There is a clear difference between men and women in terms of the presence of NSVT, which means a higher theoretical risk of SCD. Moreover, an earlier onset in men who do exercise at the semi-professional/competitive level was observed. Our group believes that it is particularly important to give these young men appropriate advice and to maintain a high degree of vigilance of arrhythmic events by carrying out regular Holter monitoring.

## Figures and Tables

**Figure 1 genes-14-00840-f001:**
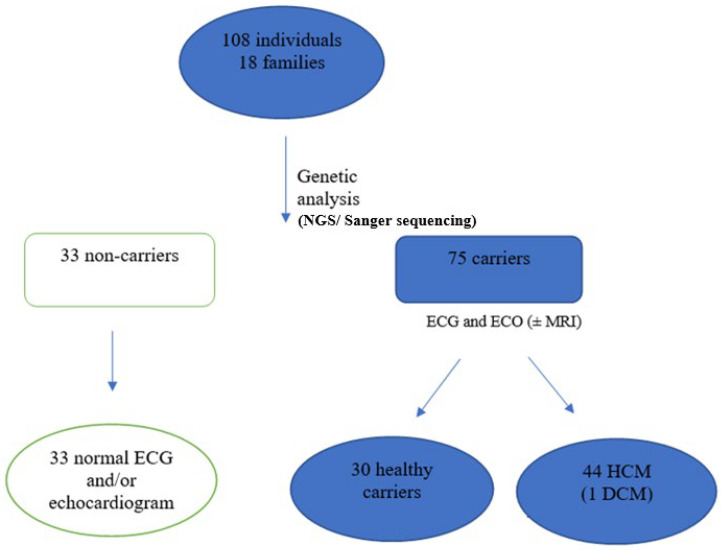
Central figure which represents the flow of participants in the genetic study and the cardiological study.

**Figure 2 genes-14-00840-f002:**
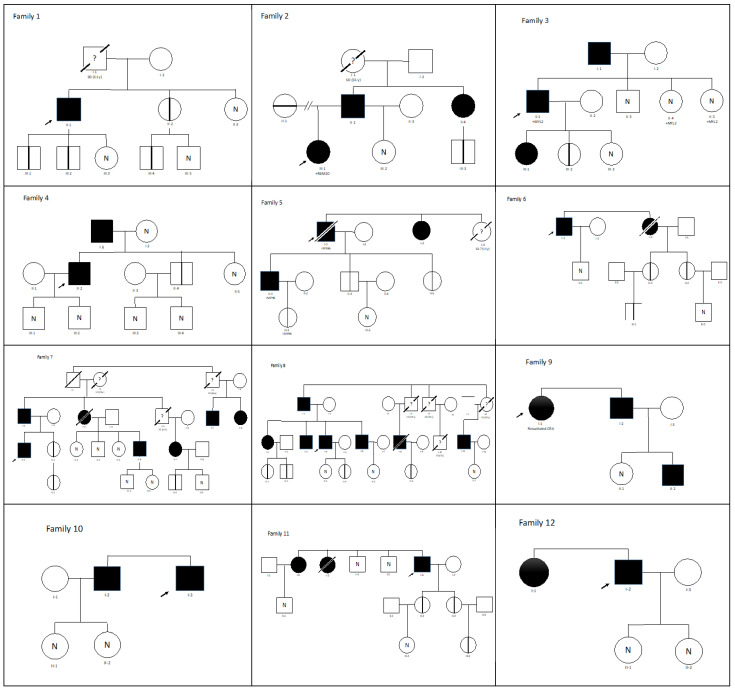
Pedigree of families with hypertrophic cardiomyopathy, carrying *MYBPC3* p.Val931Glyfs*120. SD, sudden death; SCA, sudden cardiac arrest. Symbols denote sex and disease status: box, male; circle, female; darkened, phenotype of hypertrophic cardiomyopathy; slashed, deceased; clear symbol, unaffected; without sign, not studied; N, noncarriers; vertical line, carrier without phenotype of hypertrophic cardiomyopathy; ?, unknown phenotype. Index cases signaled by black arrows.

**Figure 3 genes-14-00840-f003:**
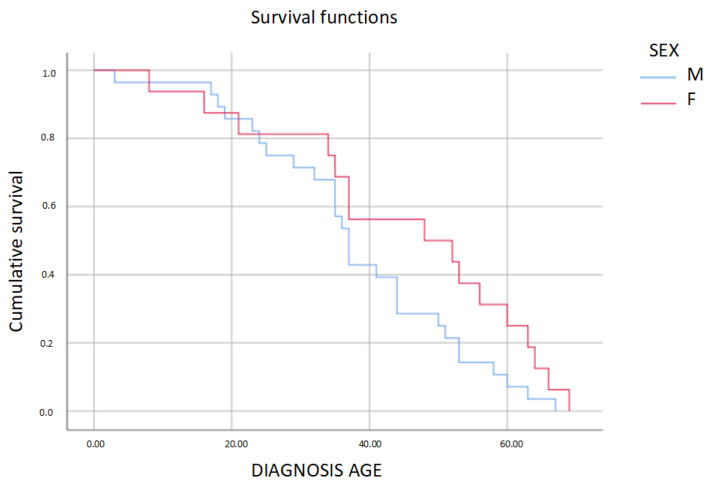
Log-rank test: penetrance of HCM according to age and sex in carriers of the p.Val931Glyfs* *MYBPC3* mutation. In our sample, there is no statistically significant difference in the average diagnosis age of men and women (*p* > 0.05).

**Figure 4 genes-14-00840-f004:**
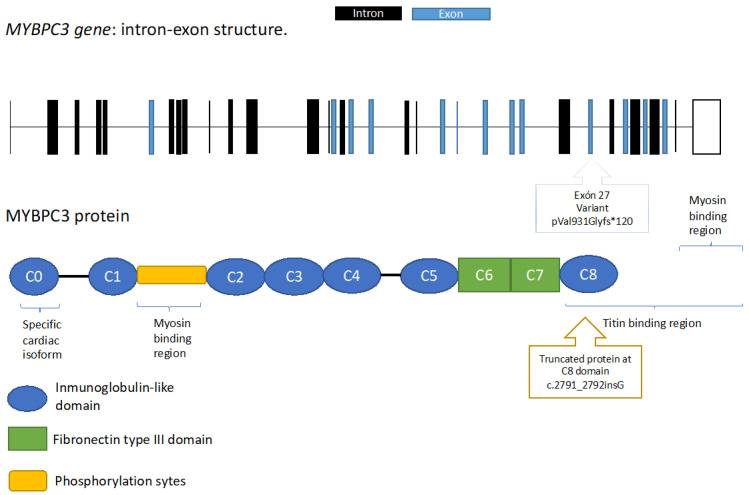
Representation of the variant p.Val931Gly*fs in gene *MYBPC3* and the truncated protein.

**Table 1 genes-14-00840-t001:** Characteristics of the 44 affected hypertrophic cardiomyopathy carriers of *MYBPC3* p.Val931Glyfs*120. Unless otherwise indicated, data are expressed as: Nº (%) or Yes/No (% Yes) or mean ± standard deviation 1. ^+^ Deceased patients are excluded from the calculation of the average age.

	Carrying Patients with HCM			
	All	Men	Women	*p*
N	44	28	16	
Current age ^+^	55 ± 17	56 ± 16	55 ± 20	0.846
Diagnosis age	41 ± 17	39 ± 15	46 ± 19	0.206
Reason for diagnosis:				0.382
Symptoms	20 (45.5%)	14 (50%)	6 (37.5%)	
Family history of disease	19 (43.2%)	10 (35.7%)	9 (56.3%)	
Irregular ECG	5 (11.4%)	4 (14.3%)	1 (6.3%)	
Thickness of left ventricle wall (IVS)	20.5 ± 5.8 (Range 13.0–43.0)	21.4 ± 6.3 (Range 13.0–43.0)	18.9 ± 4.6 (Range 13.0–26.0)	0.208
Thickness of posterior wall (LVPW)	12.2 ± 4.5 (Range 6.6–30.0)	12.3 ± 4.8 (Range 7.6–30.0)	11.8 ± 3.9 (Range 6.6–20.0)	0.759
Hypertrophy type:				0.346
Apical	1 (2.3%)	0 (0%)	1 (6.3%)	
Asymmetric septal	31 (70.5%)	21 (75%)	10 (62.5%)	
Concentric	12 (27.3%)	7 (25%)	5 (31.3%)	
Irregular ECG	37/7 (84.1%)	24/4 (85.7%)	13/3 (81.2%)	0.3524
Symptoms:	24/20 (54.5%)	16/12 (57.1%)	8/8 (50%)	0.647
Breathlessness	19(43.2%)	14 (50%)	5 (31.3%)	0.227
Chest pain	6 (13.6%)	3 (10.7%)	3 (18.8%)	0.455
Palpitations	6 (13.6%)	3 (10.7%)	3 (18.8%)	0.455
Syncope	5 (11.4%)	4 (14.3%)	1 (6.3%)	0.419
NYHA class				0.897
I + II	33 (75%)	20 (71.4%)	13 (812%)	
III + IV	11 (25%)	8 (28.6%)	3 (18.7%)	
Obstruction	11/33 (25%)	7/21 (25%)	4/12 (25%)	1
Systolic dysfunction	7/37 (15.9%)	6/22 (21.4%)	1/15 (6.3%)	0.185
Diastolic dysfunction	33/11 (75%)	23/5 (82.1%)	10/6 (62.5%)	0.148
Left atrial (LA) diameter				
Normal < 45 mm	22 (50%)	15 (53.6%)	7 (43.8%)	
Dilated ≥ 45 mm	22 (50%)	13 (46.4%)	9 (56.2%)	0.531
Atrial fibrillation (AF)	15/29 (34.1%)	8/20 (28.6%)	7/9 (43.8%)	0.307
Arrhythmia	27/17 (61.4%)	17/11 (60.7%)	10/6 (62.5%)	0.907
Non-sustained tachycardia in Holter monitoring (NSVT)	12/32 (27.3%)	11/17 (39.3%)	1/15 (8.3%)	**0.018**
Fibrosis in magnetic resonance imaging (MRI)	16/12 (57.1%)	12/5 (70.6%)	4/7 (36.4 %)	0.074
Semi-professional sport	11/33 (25%)	11/17 (39.3%)	0/16 (0%)	**0.004**
Implantable cardioverter defibrillator (ICD)	18/26 (40.9%)	15/13 (53.6%)	3/13 (18.8%)	**0.024**
Primary prevention	14 (77.8%)	11 (73.3%)	3 (100%)	
Secondary prevention	4 (22.2%)	4 (26.7%)	0 (0%)	**0.310**
Other procedures:				
Heart transplant	2 (4.5%)	2 (7.1%)	0 (0%)	
Myomectomy	3 (6.8%)	1 (3.6%)	2 (12.5%)	
Alcohol septal ablation (ASA)	1 (2.3%)	0 (0%)	1 (6.2%)	
Total events:	9 (20.4%)	6 (21.4%)	4 (25%)	0.6056
Acute myocardial infarction (AMI)	1 (2.3%)	1 (3.6%)	0 (0%)	
Stroke	2 (4.5%)	1 (3.6%)	1 (6.2%)	
Cardiac respiratory arrest (CRA)	2 (4.5%)	1(3.6%)	1 (6.2%)	
Syncope, ventricular fibrillation (FV)	1 (2.3%)	1 (3.6%)	0 (0%)	
Death by cardiac heart failure (CHF)	3 (6.8%)	2 (7.2%)	1 (6.2%)	

## Data Availability

The data presented in this study are available on request from the corresponding author. The data are not publicly available due to the protection of the privacy of the patients involved.

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
