# Peer review of "Description of a Cohort with a New Truncating MYBPC3 Variant for Hypertrophic Cardiomyopathy in Northern Spain"

_genes, 2023, doi:10.3390/genes14040840_

Round 1

Reviewer 1 Report

Description of a cohort with a new truncating MYBPC3 mutation for hypertrophic cardiomyopathy with a founder effect in northern Spain.

Fernández Suárez et al. has performed the genotype-phenotype correlation of the p.Val931Glyfs*120 variants in MYBPC3 variant in 108 individuals from 18 non-related families associated with hypertrophic cardiomyopathy (HCM). The analysis is important as it provides the understanding of the penetrance and expressivity of the variant which is one of the issues in the field. Also it provided other useful information like increase in the HCM with competitive sport in the men. They compared the symptoms, diagnosis, clinical phenotypes between the men and women. This study could be useful for the field. I would advise the author if they can make the following changes: 

Major Revision:

  1. For the extended family member please provide the mutation result like Sanger in the supplementary to confirm the results. If it is a targeted NGS panel the author can provide the IGV result.

  2. How does the author identify the first 18 index cases that has p.Val931Glyfs*120 MYBPC3 gene mutation after which they recruited 108 extended family members?

  3. How does the figure 3 justify the statement on Line 144 and 145 “The penetrance of the disease increases with age, meaning that 50% of males have developed HCM by the time they reach 55.5 years of age and 50% of women by the time they reach 60 years of age”. Please explain. 

  4. I appreciate that the author has made a table with detailed clinical information for both sexes. It would be wonderful that they can have the graphical representation with the graph that would make it more interactive and would be much helpful to understand. Also try to put the significance value in the graph.

  5. It is interesting and could be important that high level sport could increase the chances of HCM in genetically predisposed subjects. It would be helpful if the author can elaborate the mechanism, and previous study has been done on this subject in the Discussion especially with MYBPC3 gene. 

  6. It is interesting the p.Val932Glyfs*120 MYBPC3 is a founder variant. I understand that this is a clinical paper. If it is possible it would be very interesting if authors can perform an analysis where they can identify the age of the mutation and from which population the variant came to the Pasiego population. This would help to identify why this variant turns into the founder variant and spread in the population. 

Minor Revision:

  1. Figure 1 since the background does not have any meaning. Please remove it. It would make the figure better.

  2. Figure 1 there is no need to write “whose phenotype has never been published until now”. Please write till “in Cantabria”.

  3. Line 115 please remove “just”.

  4. Means mention the time of the study at line 138. 

  5. Please provide the full form of the abbreviation in Table1.

  6. Please explain the negative phenotype in the manuscript. It could be confusing.

  7. Line 276, please change “ac” to “can”.

  8. In line 276, Please change “According our study”  to “According to our study”

  9. In line 277, Please change “nnorthern” to “northern”.

  10. Please remove “>” from Line 278.

Author Response

Thank you very much for your comments and suggestions.
We really appreciate your comments, which have improved and enriched the manuscript.
Reviewer 1's comments are attached. Comments have been answered point by point. Changes in the revised manuscript have been marked using the "Track Changes”, so editors and reviewers can easily track changes.

Reviewer 2 Report

The manuscript of  Fernández Suárez et al. is focusing on a very up-to-date topic of variant penetrance and their interpretation related to this issue. The authors gathered a large cohort of 75 subjects from 18 different families from northern Spain carrying MYBPC3 p.Val931Glyfs*120 truncating variant. Some of the families are large and three-generational. The authors analyze systematically phenotypic features related to hypertrophic cardiomyopathy,  try to find genetic-phenotypic correlations and finally to give some recommendations.

Major comments:

The variant- MYBPC3 p.Val931Glyfs*120- it is true that was first mentioned in  Gómez et al. (2014), however, it was not in detail described there. Is the variant mentioned in any databases? I could not identify it in ClinVar. What is the population frequency of the variant? Have any studies been performed in Cantabria on a healthy population?

Founder effect-have any studies been performed and showed an actual block? If not, and a suspected founder effect is based only on the facts mentioned in the discussion I would leave in only for the discussion and not write as so in the title and abstract.

Methodology-what was the actual gene panel (which genes were included)? I assume that a proband got a genetic testing and other family members were just segregated for this variant or did they all get panel too?

How often do the unaffected carriers follow up?

Nomenclature-please use also an HGVS nomenclature to describe the variants, use the term variant (P/LP) instead of mutations etc.

Minor comments:

Graphical abstract - the background is too colorful and therfore ditracting

Introduction-The study of Mendez et al 2021 is mentioned. Please name a variant, because at first look it can be confusing that the same variant as in the main manuscript is meant. 

Reference 11 does not seem to be up to date (2001)

Results Line 118-121 - additional variants are mentioned. It would be useful to provide a Supplementary Table with them.

Line 129 - 70 and 77 does not seem a relatively early age. What is the life expectancy in Cantabria/Spain?

For the Results section, especially lines 177-2015 I would try to shorten it and leave only the most relevant facts.

Discussion-a Figure with pathogenic variant localization in the gene could be useful 

It would be interesting to elaborate more on the penetrance issue in the discussion and to give some examples.

Please include limitations in the Discussion

Author Response

Thank you very much for your comments and suggestions. We really appreciate your comments, which have improved and enriched the manuscript. The reviewer's comments are attached. Comments have been answered point by point. Changes in the revised manuscript have been marked using the "Track Changes”, so editors and reviewers can easily track changes.

Round 2

Reviewer 2 Report

Thank you for the revisions. 

I have just a few small comments:

1. Please include in the limitations that the variant frequency has not been tested in a local population

2. There are a few minor editorial issues: e.g. p.Val931Glyfs*120 330 MYBPC3 variant name should be the same in both places at the graphical abstract, lines 162-181 formatting, limitations are usually before conclusions etc.

3. Please consider submitting the variant to one of the public databases (especially in case required by the journal's internal rules)

Author Response

Thank you very much for your inputs. We already correct the manuscript according your suggestions.

Please include in the limitations that the variant frequency has not been tested in a local population.

We really appreciate your recommendation. As we explained in the Discussion, the penetrance data are applicable to our sample but not to a broader population, since the variant frequency has not been tested in a local population. Following the suggestion, we have included this limitation in the limitations section (lines 486 and 487).

There are a few minor editorial issues: a) p.Val931Glyfs*120 330 MYBPC3 variant name should be the same in both places at the graphical abstract, b) lines 162-181 formatting, c) limitations are usually before conclusions, etc.

Thank you very much for your advices.

  1. We have named the variant in the same way in both text boxes of the graphical abstract, including gene name MYBPC3.
  2. We have corrected the format error between lines 162 and 181.
  3. Following your recommendation, we have changed the order of the sections, placing the Limitations section before the Conclusions section

Please consider submitting the variant to one of the public databases (especially in case required by the journal´s internal rules).

Indeed, as indicated by the reviewer, we already notified this new truncating MYBPC3 variant, waiting for the publication references in Genes to include the clinical data.
